# Targets of complement-fixing antibodies in protective immunity against malaria in children

Linda Reiling[1], Michelle J. Boyle[1], Michael T. White[2], Danny W. Wilson[1,3], Gaoqian Feng[1,4], Rupert Weaver[1], D. Herbert Opi[1,5], Kristina E.M. Persson[6], Jack S. Richards[1,4], Peter M. Siba[7], Freya J.I. Fowkes[1,4,5], Eizo Takashima[8], Takafumi Tsuboi[8], Ivo Mueller [2,7,9] & James G. Beeson[1,4,5]

Antibodies against *P. falciparum* merozoites fix complement to inhibit blood-stage replication in naturally-acquired and vaccine-induced immunity; however, specific targets of these functional antibodies and their importance in protective immunity are unknown. Among malaria-exposed individuals, we show that complement-fixing antibodies to merozoites are more strongly correlated with protective immunity than antibodies that inhibit growth quantified using the current reference assay for merozoite vaccine evaluation. We identify merozoite targets of complement-fixing antibodies and identify antigen-specific complement-fixing antibodies that are strongly associated with protection from malaria in a longitudinal study of children. Using statistical modelling, combining three different antigens targeted by complement-fixing antibodies could increase the potential protective effect to over 95%, and we identify antigens that were common in the most protective combinations. Our findings support antibody-complement interactions against merozoite antigens as important anti-malaria immune mechanisms, and identify specific merozoite antigens for further evaluation as vaccine candidates.

[1] Burnet Institute, Melbourne 3004 Victoria, Australia. [2] Institute Pasteur, Paris 75015, France. [3] Research Centre for Infectious Diseases, School of Biological Sciences, University of Adelaide, Adelaide 5005, Australia. [4] Department of Medicine (Royal Melbourne Hospital) and Melbourne School of Population and Global Health, University of Melbourne, Victoria 3010, Australia. [5] Central Clinical School (Infectious Diseases; Immunology; Epidemiology and Preventative Medicine) and Department of Microbiology, Monash University, Victoria 3800, Australia. [6] Department of Laboratory Medicine, Lund University, Skåne University Hospital, 22185 Lund, Sweden. [7] Papua New Guinea Institute of Medical Research, Goroka EHP, Papua New Guinea. [8] Division of Malaria Research, Proteo-Science Center, Ehime University, Matsuyama 790-8577, Japan. [9] Walter and Eliza Hall Institute, Parkville 3050, Australia. Correspondence and requests for materials should be addressed to J.G.B. (email: beeson@burnet.edu.au)

Despite gains made through increased control efforts, malaria remains a significant health and economic burden globally, and progress in reducing the malaria burden is stalling in recent years[1]. The RTS,S/AS01 subunit vaccine targeting the *P. falciparum* pre-erythrocytic stage is now entering phase 4 implementation trials; however, with low vaccine efficacy of 18.3–36.3% depending on age and vaccine regimen, it is clear that second generation malaria vaccines will be needed[2]. In naturally acquired immunity to malaria, antibodies targeting merozoites during blood-stage infection play important roles in protective immunity, as demonstrated by several lines of evidence[3–8]. As such, targeting merozoite antigens is a key strategy of vaccines aimed at limiting parasite replication and parasite burden, thereby preventing clinical disease[5]. However, a limited understanding of key protective antigens and mechanisms of antibody-mediated protection hampers the identification and prioritization of specific merozoite antigens and combinations as vaccine candidates. Establishing of correlates of protection is a priority topic in the Malaria Vaccine Technology Roadmap[9]. Current in vitro immunoassays have proven inconsistent or insensitive as correlates of immunity in vaccine or population studies, adding to the difficulties in advancing vaccine antigens[8]. Antibodies quantified by ELISA do not consistently correlate with protective immunity from malaria and often they do not reflect the functional activity of antibodies[10–17]. The current standard assay for quantifying functional activity of merozoite antigen vaccines, the growth inhibition assay (GIA), quantifies the ability of antibodies to inhibit parasite replication in vitro. While activity in GIA has shown predictive value in some pre-clinical animal models, antibody activity in GIAs has not reliably correlated with protective immunity in studies of naturally acquired immunity or in clinical vaccine trials[10,13,16,18–23].

Recent studies demonstrated that acquired and vaccine-induced human antibodies to merozoite antigens can fix and activate serum complement, leading to inhibition of RBC invasion and merozoite death[24,25]. It was found that a large proportion of naturally acquired human antibodies only effectively prevented merozoite invasion in the presence of complement[24]. A role for complement is further supported by associations between protection from malaria and levels of cytophilic IgG1 and IgG3, which can activate the classical complement cascade by binding C1q[26–30]. This important role of complement in human antibody function may explain why standard in vitro GIAs, which are performed in complement-free conditions, are neither strongly nor consistently associated with protection[10,13,21,31,32]. Given these assays are still used as a gold standard to assess and prioritize vaccine candidates, important targets of functional antibodies might be missed[19,33,34]. Roles for antibody-complement interactions have also been recently reported for immunity to sporozoites[35–37], suggesting that antibody-complement interactions have wider roles in immunity against multiple stages of malaria.

Specific antigen targets mediating complement-dependent protection are currently unknown. Initial studies demonstrated that antibodies to MSP1 and MSP2 could mediate complement-dependent invasion-inhibitory activity in vitro[24], but the roles of these target-specific complement-dependent antibodies in protective immunity are yet to be explored, and other potential candidates have not been assessed. Although recent gains have been made in defining mechanisms of immunity to malaria[8], there is a paucity of data on protective associations for a range of antigen-specific functional responses in human studies; to date, studies assessing responses to multiple merozoite antigens have only assessed IgG reactivity using standard immunoassays, such as ELISA and protein microarrays[31,38,39]. Furthermore, there is a lack of validated and practical in vitro assays to quantify functional antibody responses to a range of individual antigens in studies of naturally-acquired immunity in populations where antibodies with a broad range of specificities are acquired.

The prior finding that complement-fixing antibodies to whole intact merozoites were highly correlated with protection from clinical malaria and high-density parasitemia in children[24] provided a strong rationale to identify specific antigen targets of these antibodies that are linked with protective immunity. Here, we compare protective associations for complement-fixing antibodies with the current reference functional assay, GIA. We develop optimized assays that quantify complement-fixing antibodies to individual merozoite surface proteins and define the acquisition of complement-fixing antibodies to a range of merozoite antigens. In a longitudinal cohort study of children, we apply our novel strategy to identify specific antigens that are targets of complement-fixing antibodies associated with protection. Using statistical models, we evaluate protective associations for all possible combinations of responses and identify antigen-specific complement-fixing antibody combinations that give the strongest protective associations. These findings will help enable potential vaccine candidates to be selected and prioritised based on functional activity.

## Results

**Complement-fixing antibodies more strongly associated with protection than GIA.** We previously determined that complement-fixing antibodies to whole merozoites were associated with protection against clinical malaria in a longitudinal cohort of children in a malaria-endemic region of Papua New Guinea (PNG)[24]. GIA is currently the standard assay used to evaluate functional antibodies to merozoites and, therefore, we evaluated GIA activity among the same children's cohort and compared their relationship with protection from malaria to that for complement-fixing antibodies (which was assessed as fixation of the first component of the classical complement cascade, C1q, to the surface of merozoites[24]). The Cox proportional hazards model was used to determine the hazard ratio (HR) for prospective risk of malaria over 6 months follow-up among children with different levels of complement-fixing and GIA functional antibodies, with adjustment (adjusted HR, aHR) for established confounders of age and location of residence[24]. In contrast to the strong association between complement-fixing antibodies with protection from clinical malaria (high vs low responders aHR = 0.15 (95% CI 0.06–0.35, $p < 0.0001$), inhibitory activity quantified by standard GIA was only weakly associated with protective immunity (high vs low responders aHR = 0.60 (95% CI, 0.33–1.09; $p = 0.094$) (Fig. 1a; Supplementary Fig. 1) confirming a greater protective association of complement-fixing antibodies. The protective association ((1-HR) x 100%) of complement-fixing antibodies was 45% higher (absolute difference) than growth-inhibitory antibodies.

To investigate relationships between antibody parameters further, we tested a selection of cohort samples in antibody-complement invasion inhibition assays (AbC-IIA) using purified merozoites[24]. Invasion in the presence of antibodies and complement, was negatively correlated with C1q-fixation to the merozoite surface ($r = -0.62$, 95% CI $-0.81$–0.30, $p = 0.0006$ [Spearman's rho], Fig. 1b), consistent with our prior conclusion that complement is important for inhibitory activity of antibodies[24]. In contrast, we observed no significant correlation between antibody-mediated C1q-fixation and activity in standard GIA, which is performed in complement-free conditions ($r = -0.29$, 95% CI $-0.61$–0.12, $p = 0.15$ Fig. 1c). There was some correlation between activity in AbC-IIA and GIA (both measured as invasion/growth, $r = 0.4$, 95% CI 0.02–0.69, $p = 0.035$, Fig. 1d);

as noted previously, some antibodies can act to inhibit invasion independent of complement[24]. We also observed a significant difference in AbC-IIA functional activity between groups when cohort participants were stratified into tertiles according to low, medium and high C1q fixation to the merozoite's surface. There was higher median invasion (less inhibition) in the group of low C1q fixation compared to medium and high C1q fixation (Fig. 1e). However, there were no significant differences in activity in GIAs between the groups (Fig. 1f).

**Development of an antigen-specific complement-fixation assay.** Given the strong protective association for antibody-complement-fixation on the merozoite surface, we aimed to identify specific merozoite antigens that were targets of potentially protective, complement-fixing antibodies. We first developed a plate-based assay to detect and quantify complement-fixing antibodies to individual merozoite antigens. Antigens were coated onto plates, incubated with individual, heat-inactivated (complement-inactive) serum samples from a cohort of children as a source of antibodies, followed by incubation with purified C1q (pC1q). C1q-fixation was detected as a measure of complement-fixing activity of antibodies. To validate pC1q, a selection of PNG adult's and children's serum samples from a cross-sectional cohort[40] was tested using pC1q versus normal serum (NS, complement active) as the source of C1q, on the major merozoite antigen MSP2 (FC27-allele). There was a strong positive correlation between assays with pC1q versus normal serum ($r = 0.95$, $p < 0.0001$ [Spearman's rho]) (Supplementary Fig. 2). The use of pC1q was favoured for subsequent assays because it facilitated the use of standardized reagents.

We demonstrated that fixation of C1q by antibodies resulted in the activation of the entire complement cascade by investigating the correlation between C1q fixation and the formation of C5b-C9 which forms the membrane attack complex (MAC), using selected merozoite antigens as antibody targets and adult and children serum samples from a cross-sectional study in PNG. C1q-fixation on EBA175RII, MSP2 (FC27) and MSP-DBL1 correlated strongly with levels of MAC formation on the respective antigen, using normal serum as a complement source ($r = 0.82$–$0.98$, all $p < 0.0001$, Fig. 2a). To confirm that MAC formation resulted from fixation of C1q, we tested antibody-mediated complement fixation using either normal serum, heat-inactivated serum or C1q-depleted serum as the source of complement factors. C1q-fixation and MAC formation were greatly reduced in heat-inactivated serum or C1q-depleted serum compared to normal serum (Fig. 2b and Supplementary Fig. 3A). Furthermore, we tested a human monoclonal antibody (mAb) against MSP2 (FC27) that had two leucine-to-alanine conversions (L234A and L235A, LALA mutant); this mutation in the Fc region results in reduced C1q binding[41]. While both wild type and mutant mAbs bound to MSP2 to the same extent (Supplementary Fig. 3B), the ability of the mutant mAb to bind and fix C1q, leading to activation of the complement cascade and MAC formation, was greatly reduced compared to wild-type mAb for C1q fixation and MAC formation (>10-fold and >30-fold, respectively). The mAbs were tested at a concentration (5 μg/ml) that reached maximum detection of MAC formation for wild type (Fig. 2c, d).

**Acquisition of complement-fixing antibodies in children.** We next examined the prevalence and magnitude of C1q-binding antibodies against a range of merozoite antigens in a longitudinal cohort of PNG children[42]. Antigens included were selected on the basis of pre-screening using samples from a cross-sectional study including malaria-exposed samples from adults in the same

population[43] (representative examples are shown in Supplementary Fig. 4). Results from this screen showed that C1q-fixation was variable amongst different merozoite antigens, and not all high IgG responders to a given antigen had also high levels of C1q fixation. Antigens with no or very low detectable C1q-fixation in the pre-screen were not included in the longitudinal cohort; excluded antigens were PfRH4.9, Pf12, Pf38, Pf41 and SERA5. A total of $n = 22$ antigens were tested for complement-fixing antibodies in the longitudinal cohort. The tested antigens comprised established and emerging vaccine candidates: MSP1–19, MSP1–42, MSP2 (3D7 and FC27 alleles), MSP3, MSP4, MSP6, MSP7, MSP9, MSP10, MSPDBL1, AMA1, EBA140 region II (RII), EBA140 region III-V (RIII-V), EBA175 RII, EBA175 RIII-V, PfRH2-2030, PfRH5, Ripr, GAMA, RALP1 and Pf113.

C1q-fixing antibodies in the longitudinal cohort were detected in the majority of children for each recombinant antigen tested. The prevalence of C1q-fixing antibodies to most antigens was greater than 90% (13 out of 22; MSP1–19, MSP1–42, MSP2 (3D7), MSP4, MSP6, MSP7, MSP9, AMA1, EBA140RII, EBA140RIII-V, EBA175 RII, RH2-2030 and GAMA, Supplementary Table 1). The targets with the lowest prevalence of C1q-fixing antibodies were MSP3 (31.8%), PfRH5 (57.4%), Pf113 (60.3%) and MSP10 (64.1%).

For the majority of antigens tested, the level and prevalence of C1q-fixing antibodies was higher among older children, consistent with the acquisition of functional antibodies with age and cumulative exposure (Table 1 and Supplementary Table 1). Complement-fixing antibodies were also higher in magnitude and prevalence in children with *P. falciparum* parasitemia at the time of sample collection, consistent with boosting of functional antibodies by active infection.

C1q-fixing activity to most merozoite proteins were strongly correlated with C1q-fixing activity to the intact merozoite surface (Fig. 3; representative examples shown in Supplementary Fig. 5; range from $r = 0.29$, 95% CI 0.15–0.41 (Ripr) to $r = 0.80$, 95% CI 0.74–0.84 (EBA140RIII-V), 9/22 comparisons $r > 0.6$ and 7/22 comparisons $r > 0.7$; all comparisons $p < 0.0001$ [Spearman's rho]). Complement-fixing antibodies to individual surface proteins were generally moderately ($r > 0.5$) to highly ($r > 0.6$) positively correlated, suggesting co-acquisition of complement-fixing antibodies to multiple targets from malaria infections (Fig. 3). However, low-level correlations were generally observed for responses to MSP3, Ripr and Pf113 and all other antigens, respectively. Complement-fixing antibodies to the two different alleles of MSP2 (3D7 and FC27) were only moderately correlated ($r = 0.506$, $p < 0.001$), consistent with findings that naturally acquired antibodies largely target allele-specific epitopes[44]. The correlation coefficients between total IgG reactivity and C1q-fixation to an antigen were variable among antigens with the weakest correlations observed for MSP3 ($r = 0.29$, 95% CI 0.16–0.42), Ripr ($r = 0.46$, 95% CI 0.34–0.57), PfRH5 ($r = 0.48$, 95% CI 0.36–0.59) and MSP1–19 ($r = 0.57$, 95% CI 0.47–0.66); The strongest correlations were observed for AMA1 ($r = 0.95$, 95% CI 0.93–0.96), MSP4 ($r = 0.92$, 95% CI 0.90–0.94) and EBA175RII ($r = 0.92$, 95% CI 0.90–0.94) (Supplementary Table 2).

**Function correlates with cytophilic subclasses but weakly with avidity.** The correlations between C1q-fixing antibodies and IgG subclasses to selected merozoite antigens, namely MSP2, GAMA, AMA1 and MSP-DBL1 as representative results, were explored. IgG1 and IgG3 subclasses can effectively fix complement, whereas IgG2 and IgG4 have little or no activity[45]; therefore, correlations for IgG1 and IgG3 were evaluated. It was previously demonstrated that IgG1 and IgG3 are the dominant responses to

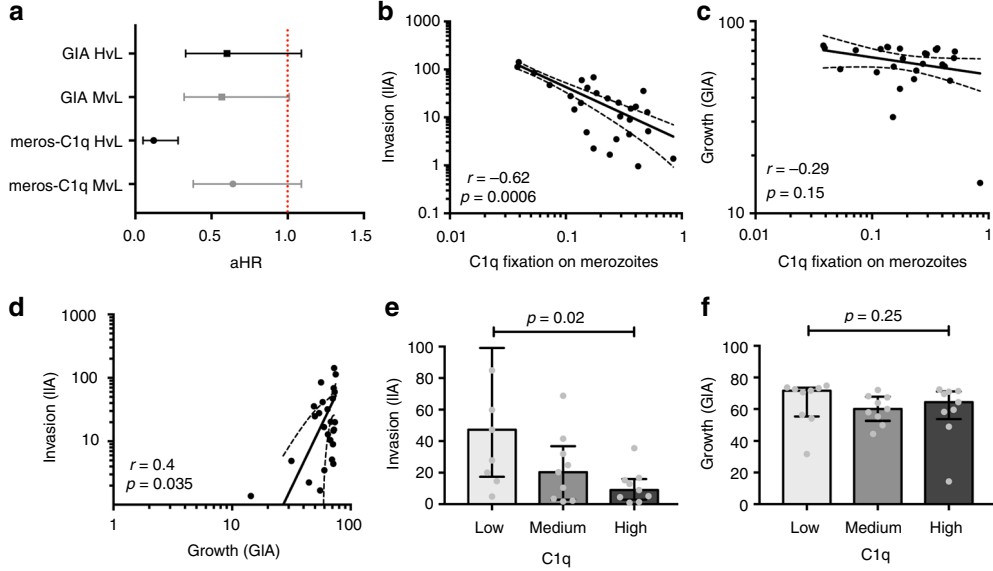

**Fig. 1** Protective associations and correlation between antibody-dependent fixation of C1q and complement-dependent invasion inhibition or GIA. **a** A longitudinal cohort of PNG children was used to calculate protective associations for functional antibodies as determined in Growth Inhibition Assays (GIA, complement-free conditions, $n = 205$, Mugil cohort) and C1q-fixation assays on the merozoite surface ($n = 200$). Children were stratified into three equal tertiles (based on ranks) and adjusted hazard ratios (aHR) were calculated using Cox proportional hazards model comparing high versus low (HvL) and medium versus low (MvL) responders. Black bars compare high versus low responders (HvL), grey bars compare medium versus low responders (MvL). **b** The correlation between C1q fixation on the merozoite surface and activity in antibody-complement invasion inhibition assays (IIA) among a subset of samples ($n = 27$) (data are shown as invasion (%) relative to controls). **c** The correlation between C1q fixation and activity in growth inhibition assays (GIAs) among the same subset of samples ($n = 27$) (data are shown as growth (%) relative to controls). **d** Shows the correlation between antibody-complement IIA and GIA among a subset of samples ($n = 27$). Spearman's rho (r) and statistical significance (p) are indicated. Non-linear regression was used to generate a trendline. Dotted lines indicate 95% confidence intervals. **e, f** Study participants ($n = 27$) were stratified into three equal tertiles according to low, medium and high C1q fixation to the merozoite surface. The median invasion measured by antibody-complement IIA (**e**) and growth measured by GIA (**f**) was compared between the three groups (statistical significance determined by Kruskal–Wallis test)

merozoite antigens in this population, with very little IgG2 and IgG4[27,29,46,47]. MSP2-specific IgG3 levels were strongly correlated with C1q-fixing antibodies to MSP2 ($r = 0.84$, 95% CI 0.79–0.88, $p < 0.0001$ [Spearman's rho]), and this correlation was weaker between IgG1 and C1q-fixing antibodies ($r = 0.39$, 95% CI 0.26–0.50, $p < 0.0001$; Fig. 4a). For AMA1, both IgG1 and IgG3 were significantly correlated with the presence of C1q-fixing antibodies, however, the IgG1 response was much stronger (Fig. 4c, $r = 0.89$, 95% CI 0.85–0.91, $p < 0.0001$, and $r = 0.60$, 95% CI 0.51–0.69, $p < 00001$, respectively). For GAMA and MSP-DBL1 we observed moderate to strong positive correlations between the presence of IgG1 and IgG3, and antibody-dependent C1q fixation (Fig. 4b, d). IgM responses to merozoite antigens may also contribute to complement fixation, and we examined correlations to several antigens to explore this. IgM reactivity was also significantly correlated with complement fixation ($r = 0.38$ for MSP2; $r = 0.69$ for EBA140II; $r = 0.53$ for EBA175III-V; $p < 0.0001$ for all [$n = 194–197$]).

Using samples from the cross-sectional cohort of adults and children, we analysed the association between C1q fixation on MSP2 and the avidity of MSP2 antibodies as measured by surface plasmon resonance (SPR). There was a weak correlation between C1q fixation and increased avidity of antibodies measured as $k_d$ against MSP2 (3D7 allele) ($r = -0.24$, $p = 0.05$), and no association between C1q fixation and avidity of antibodies against MSP2 (Fc27) ($r = -0.07$, $p = 0.56$; Supplementary Fig. 6). To explore the relative contributions of IgG magnitude and antibody avidity to complement-fixing activity, we performed linear regression analysis. The addition of avidity as a predictor variable did not substantially increase $R^2$ for the relationship between IgG and complement-fixation (0.8652 compared to 0.8613 with total

IgG as a predictor alone), further suggesting that antibody avidity levels are not a major determinant of complement fixation.

**Antigen-specific complement-fixing antibodies and protection from malaria.** To identify targets of complement-fixing antibodies that may be important for protection from malaria, we analysed associations between complement (C1q) fixation to all antigens individually and protection from clinical malaria and high-density parasitemia over 6 months of follow up in the longitudinal cohort of PNG children. Children were classified into three equal groups (low, medium and high) based on complement-fixing antibodies activity for each antigen. We analysed the relationship between antibody categories and time to clinical episode, or high-density parasitemia infection (Supplementary Tables 3–5 for additional comparisons). Hazard ratios (Cox proportional hazards) were adjusted for known confounders, age and location[42]. We observed statistically significant associations for most antigens, and the strength of protective associations varied substantially between antigens, with a range from aHR 0.20 [0.10–0.43] to 1.07 [0.72–2.35] (Fig. 5a; Supplementary Fig. 7; Supplementary Tables 3–5). The 10 antigens with the strongest association with protection were EBA140RIII-V (aHR = 0.20), MSP7 (aHR = 0.23), RALP1 (aHR = 0.24), GAMA (aHR = 0.27), PfRH2 (aHR = 0.30), MSP-DBL1 (aHR = 0.33), PfRH5 (aHR = 0.37), EBA175-RIII-V (aHR = 0.41) and MSP2-3D7 (aHR = 0.41) (Fig. 5a). Complement-fixing antibodies to MSP1–19 had a significant association with protection, which supports our previous finding that antibodies against MSP1–19 mediate complement-dependent invasion inhibition[24]. Complement-fixing antibodies to Pf113 and MSP3 were not

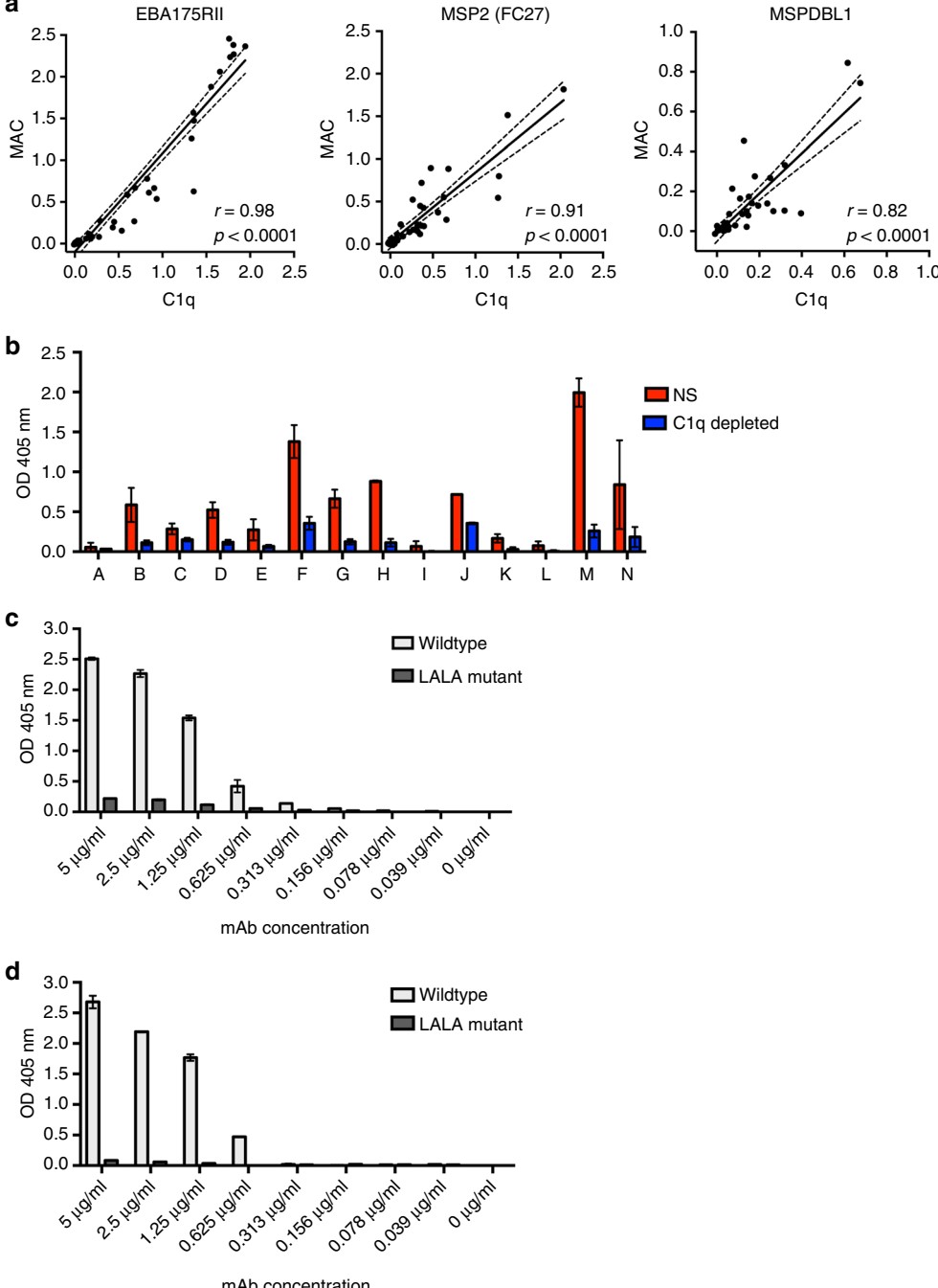

**Fig. 2** Antibody-dependent fixation of C1q and detection of downstream factors of the complement cascade. **a** Figures show the correlation between antibody-mediated fixation of C1q or membrane attack complex (MAC) among a selection of samples from the XMX cohort. Results are shown for three recombinant proteins: EBA175RII ($n = 48$), MSP2 (FC27) ($n = 43$) and MSPDBL1 ($n = 44$). Spearman rho and $p$ values are shown. Non-linear regression was used to generate a trendline, and dotted lines indicate 95% confidence intervals. Values on the x-axes and y-axes represent OD values (405 nm) for MAC reactivity or C1q activity, respectively. **b** Complement-fixing activity of antibodies was determined using a selection of samples ($n = 14$, XMX cohort) against MSP2 recombinant protein as the target. MAC formation was determined using normal serum (NS) or C1q-depleted serum in plate-based assays. Results are shown as mean ODs from 2 independent assays, performed in duplicate. Error bars represent range; x-axis: serum samples A-N. **c, d** Complement-fixation activity of a human monoclonal antibody against MSP2; the wild-type form was compared to its mutant version (LALA) that had 2 Leucine-to-Alanine conversions (L234A, L235A), which is known to inhibit C1q binding. Antibodies were tested for their ability to fix C1q (**c**) and promote MAC formation (**d**). Concentrations of antibody are indicated on the x-axis as [μg/ml]. Results are shown as the mean of two assays performed in duplicate. Error bars represent range

associated with protection; antibodies to MSP1–42, MSP4, and MSP2-FC27 showed a trend towards protective associations with aHR ranging from 0.54–0.62. Naturally acquired antibodies to MSP2 are known to be largely allele-specific (3D7 and FC27

alleles)[24,25] and infections with both alleles occurred in our study population (although 3D7 is ~2-times more common than FC27[27,29]). Therefore, we analysed associations with protection considering both allelic responses. Children were classified as

**Table 1 Magnitude of complement-fixing antibodies against merozoite antigens**

| Antigen | Age | | | Enrolment *P. falciparum* parasitemic status | | |
|---|---|---|---|---|---|---|
| | <9 yrs n[a] | ≥9 yrs n[a] | *P*[c] | PCR- n[a] | PCR+ n[a] | *P*[b] |
| Merozoites | 0.24 [0.14–0.33] | 0.35 [0.26–0.43] | <0.0001 | 0.23 [0.08–0.34] | 0.34 [0.23–0.49] | <0.0001 |
| MSP1–19 | 0.26 [0.19–0.42] | 0.29 [0.19–0.51] | 0.3 | 0.23 [0.17–0.37] | 0.29 [0.2–0.49] | 0.05 |
| MSP1–42 | 0.15 [0.04–0.42] | 0.29 [0.11–0.6] | 0.007 | 0.11 [0.02–0.33] | 0.32 [0.12–0.57] | <0.0001 |
| MSP2 (3D7) | 0.1 [0.04–0.21] | 0.164 [0.09–0.28] | 0.003 | 0.07 [0.03–0.17] | 0.17 [0.09–0.28] | <0.0001 |
| MSP2 (FC27) | 0.04 [0.009–0.15] | 0.09 [0.02–0.47] | 0.03 | 0.02 [0.001–0.1] | 0.1 [0.02–0.42] | <0.0001 |
| MSP3 | 0.24 [0.17–0.35] | 0.22 [0.15–0.34] | 0.2 | 0.19 [0.12–0.25] | 0.25 [0.18–0.37] | 0.0004 |
| MSP4 | 0.14 [0.05–0.35] | 0.22 [0.12–0.48] | 0.007 | 0.1 [0.04–0.19] | 0.25 [0.12–0.57] | <0.0001 |
| MSP6 | 0.17 [0.11–0.42] | 0.24 [0.13–0.52] | 0.19 | 0.14 [0.08–0.27] | 0.25 [0.14–0.52] | 0.0003 |
| MSP7 | 0.34 [0.17–0.58] | 0.42 [0.22–0.68] | 0.08 | 0.23 [0.11–0.46] | 0.44 [0.28–0.75] | <0.0001 |
| MSP9 | 0.08 [0.06–0.14] | 0.17 [0.08–0.35] | <0.0001 | 0.07 [0.05–0.13] | 0.16 [0.09–0.33] | <0.0001 |
| MSP10 | 0.25 [0.13–0.42] | 0.29 [0.16–0.54] | 0.2 | 0.18 [0.11–0.32] | 0.31 [0.18–0.54] | 0.0001 |
| MSP-DBL1 | 0.44 [0.1–1.06] | 0.62 [0.21–1.37] | 0.05 | 0.22 [0.07–0.94] | 0.65 [0.26–1.27] | 0.003 |
| Ripr | 0.34 [0.23–0.46] | 0.31 [0.2–0.56] | 0.86 | 0.26 [0.18–0.48] | 0.33 [0.24–0.53] | 0.03 |
| GAMA | 0.27 [0.16–0.57] | 0.43 [0.29–0.59] | 0.006 | 0.25 [0.13–0.45] | 0.44 [0.27–0.61] | <0.0001 |
| RALP1 | 0.15 [0.08–0.36] | 0.38 [0.18–0.66] | <0.0001 | 0.16 [0.09–0.33] | 0.33 [0.15–0.6] | 0.0002 |
| AMA1 | 0.26 [0.08–0.82] | 0.38 [0.18–0.78] | 0.056 | 0.16 [0.04–0.47] | 0.48 [0.22–0.96] | <0.0001 |
| EBA140 RII | 0.89 [0.44–1.25] | 1.05 [0.65–1.36] | 0.02 | 0.85 [0.29–1.3] | 1.03 [0.69–1.32] | 0.08 |
| EBA140 RIII-V | 0.1 [0.05–0.26] | 0.18 [0.08–0.39] | 0.01 | 0.09 [0.05–0.2] | 0.19 [0.09–0.46] | <0.0001 |
| EBA175RIII-V | 0.14 [0.07–0.30] | 0.26 [0.14–0.64] | 0.0006 | 0.12 [0.07–0.26] | 0.25 [0.13–0.66] | <0.0001 |
| EBA175RII | 0.70 [0.08–1.33] | 1.04 [0.28–1.43] | 0.013 | 0.28 [0.04–0.85] | 1.12 [0.33–1.43] | <0.0001 |
| Rh2-2030 | 0.26 [0.1–0.49] | 0.3 [0.14–0.50] | 0.22 | 0.16 [0.08–0.38] | 0.31 [0.17–0.56] | 0.0001 |
| PfRh5 | 0.19 [0.15–0.29] | 0.26 [0.17–0.41] | 0.01 | 0.17 [0.12–0.26] | 0.25 [0.17–0.38] | 0.0004 |
| Pf113 | 0.07 [0.04–0.12] | 0.1 [0.06–0.17] | 0.03 | 0.06 [0.03–0.12] | 0.09 [0.05–0.16] | 0.008 |

Median optical density [OD] and interquartile range [IQR] are displayed. The 3D7 reference strain was used for all antigens; the FC27 strain was also assessed for MSP2
[a]Numbers (n) analysed (n < 9 yrs, n ≥ 9 yrs, n PCR-, n PCR+) are as follows: 87, 109, 64, 132 for MSPDBL1, MSP7, GAMA, RALP1; 87, 110, 64, 133 for MSP6, EBA140RII, EBA140RIII-V, Rh5; 88, 110, 64, 133 for MSP1–42, EBA175RIII-V; 90, 110, 65, 135 for Merozoites, MSP1–19, EBA175RII, MSP2 (3D7 and FC27); 90, 111, 65, 136 for MSP4, AMA1; 90, 114, 65, 139 for Ripr; RH2-2030: 88, 110, 64, 134; MSP9: 84, 109, 63, 130; MSP10: 88, 104, 62, 130; Pf113: 85, 94, 58, 121
[b,c]*p* indicates statistical significance (Wilcoxon rank-sum tests)

high, medium, or low responders to each allele (tertiles based on ranks using C1q-fixation data), and subsequently grouped according to response levels to both alleles. Children who were high responders to both alleles had a strong protective association (aHR = 0.219 [0.06–0.73]) compared to children who were in the lower response group (which included low responders to both alleles and those who were low responders to one allele and a medium responder to the other allele). Of interest, the number of antigens with strong and intermediate associations with protection was >2-fold higher for apical proteins compared to surface proteins, suggesting that antigens residing in the apical organelles are key targets of protective complement-fixing antibodies (Fig. 5b).

In order to account for potential problems in analysis of protective associations that may arise if some children are not exposed during the period of follow-up[48], we performed a sensitivity analysis for complement fixation on whole merozoites including only those children who became re-infected (detected by PCR) over the course of the follow-up period (n = 191 compared to n = 200). This had no impact on the results (aHR all children: 0.15 [0.06–0.35, p < 0.0001; aHR re-infected children only: 0.14 [0.06–0.33], p < 0.0001).

**Breadth of complement-fixing antibodies predicts protection.** To investigate the breadth of complement-fixing antibodies and protective immunity, we assigned each participant in our cohort a breadth score based on the level of reactivity to each of the 22 antigens included in the study. Cox regression analysis showed a strong protective association against clinical malaria among children with a high breadth score compared with a low breadth score (aHR 0.27, 95% CI 0.13–0.54, p < 0.0001 [Cox proportional hazards]). There was significantly increased breadth score in older children compared to younger children, and in children who were parasitemic at enrolment compared to aparasitemic children (p ≤

0.01; Fig. 5c, d, see Supplementary Fig. 8 for breadth score distributions), consistent with acquisition of a repertoire of functional antibodies with increasing age and exposure. Furthermore, the breadth score was significantly higher in children that were protected and did not experience any clinical episodes during follow up compared to children who experienced one malaria episode (susceptible 1) or 2 or more malaria episodes (susceptible 2, p = 0.0001, Fig. 5e [Kruskal–Wallis test]).

**Combinations of antibody responses and protection from malaria.** We investigated the protective associations of combined responses to multiple antigens on to understand how protection against malaria relates to functional antibodies to multiple antigens, and to identify antigen combinations with the highest protective associations that may inform vaccine development. We evaluated potential protective efficacy (PPE), which was calculated as (1-Hazard Ratio)*100, where 100% represents complete protection. We included responses against 20 different antigens (all antigens except MSP1–42 since it is part of the same antigen as MSP1–19, and only the 3D7 and not FC27 allele of MSP2). Antigens were grouped into all possible combinations of responses of 2, 3, 4, 5, or 6 antigens (190, 1140, 4845, 15504, and 38760 combinations, respectively; n = 20 antigens were included).

We found that the PPE and the frequency of highly protective combinations (PPE >75%) increased with increasing breadth of responses (Fig. 6a). PPE increased substantially up to combinations of 3 antigens; increasing the combination size to 4–6 antigens gave only minor improvements in PPE. Interestingly, the PPE for the best combination of three antigens was very high (93.6% [95% CI 81.9–97.7]) and similar to the best combinations of 4, 5 or 6 antigens (Fig. 6b). We then ranked the PPE of all possible combinations of a given size, and calculated the median PPE of the top 100 combinations for each combination size (Fig. 6c). The median PPE of the top 100 combinations in each

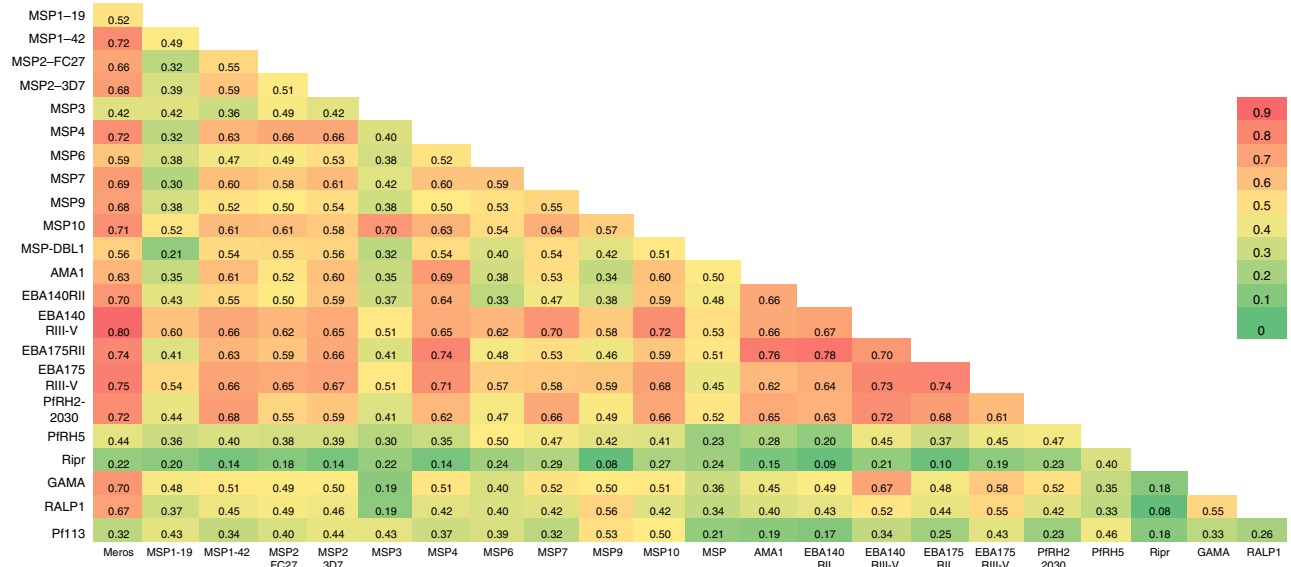

**Fig. 3** Correlations between complement-fixing antibodies to different antigens. Spearman's rho values indicate the level of correlation between C1q-fixing antibodies against different merozoite antigens in children's serum samples ($n = 196–201$). Correlates are coloured from red (highest level of correlation) to green (lowest level of correlation). All correlations were significant except for comparisons between Ripr and MSP1–42, MSP2(3D7), MSP4, MSP9, AMA1, EBA140RII, EBA175RII and RALP1, respectively ($p > 0.05$)

group rose with increasing combination size; however, this was most striking up to combinations of three antigens and the gains in PPE were modest when combination size was increased to four or more (median PPE was 88.9, 92.9, 94.6 and 95.5% for combinations of 3, 4, 5 and 6 antigens, respectively). Several combinations had PPE >95% when three or more responses were combined. Together, these findings suggest that an optimal combination of three antigens might be sufficient for high protective efficacy, or to act as a correlate of protection, and larger combinations of antigens may offer limited benefit.

To determine which specific antigens in combinations may be contributing most to PPE, we ranked all combinations based on their PPE value and identified the top 1% of combinations with the highest PPE; this was performed for each combination size (2, 3, 4, 5 and 6 antigens). We then then determined the frequency at which a specific antigen was present in the top 1% PPE combinations for each combination size (Table 2). For example, among the highest-ranked 3-antigen combinations, RALP1, MSP7 and Ripr, featured most commonly in the most protective combinations (present in 100%, 72.7% and 36.4% of the top 1% of combinations, respectively). EBA140-RIII-V, PfRH2 and MSP-DBL1 responses were also common in the most protective combinations (all 18.2%). Considering all combination sizes, these antigens were generally frequent in the most protective combinations. Interestingly, in the combinations of 5 and 6 antigens, PfRH5 (31.6% and 44.4%, respectively) and EBA175-RII (32.3% and 35.1%, respectively) became more frequently present.

## Discussion

Merozoite surface antigens have been the focus of much research as important targets of human immunity and potential vaccine candidates. However, identifying, validating and prioritizing candidates has been challenging due to incomplete understanding of mechanisms of immunity and targets of protective responses[49], and lack of methodologies that enable antigen-specific functional antibodies to be systematically evaluated. We demonstrated that complement-fixing antibodies to the intact merozoite surface are more strongly associated with protective immunity than antibody

activity in standard GIA, which is the current reference assay for evaluating merozoite vaccine antigens. Subsequently, we developed and validated an approach to quantify complement-fixing antibodies to individual merozoite antigens for identifying potential targets of protective complement-fixing antibodies. Using this approach we identified several antigens with complement-fixing antibodies that were very strongly associated with protection from malaria, supporting their potential role as immune targets and vaccine candidates and providing data supporting ongoing development of several emerging vaccine candidates. Combining complement-fixing antibody responses against as few as three specific antigens the PPE could be increased to >95%, which supports the concept of a multi-antigen vaccine approach.

Identifying and developing assays to assess immunologic correlates of protection is priority area in the WHO Malaria Vaccine Technology Roadmap[9]. Recent findings that human antibodies interact with serum complement to inhibit merozoite invasion and blood-stage replication provided the rationale for this study to assess the importance of complement-fixing antibodies as immune correlates relative to current widely used assays, and to identify targets of complement-fixing human antibodies associated with protective immunity. The low protective association for antibodies quantified in GIA, which is performed in complement-free conditions, is consistent with our previous finding that the majority of acquired antibodies require complement interactions to effectively inhibit merozoite invasion[24]. These findings have significant implications for evaluation of vaccine candidates using GIA. While this assay maybe suitable for a small number of antigen candidates where antibodies function by direct blocking activity, our results strongly suggest that complement-fixing antibodies are important for a wide range of merozoite targets. Therefore complement-fixing assays should be considered alongside other assays in vaccine development studies and trials.

To identify specific antigens that were targeted by complement-fixing antibodies we established an efficient plate-based assay to quantify complement-fixing antibodies. We based this on fixation of C1q, the first step in the classical (antibody-mediated)

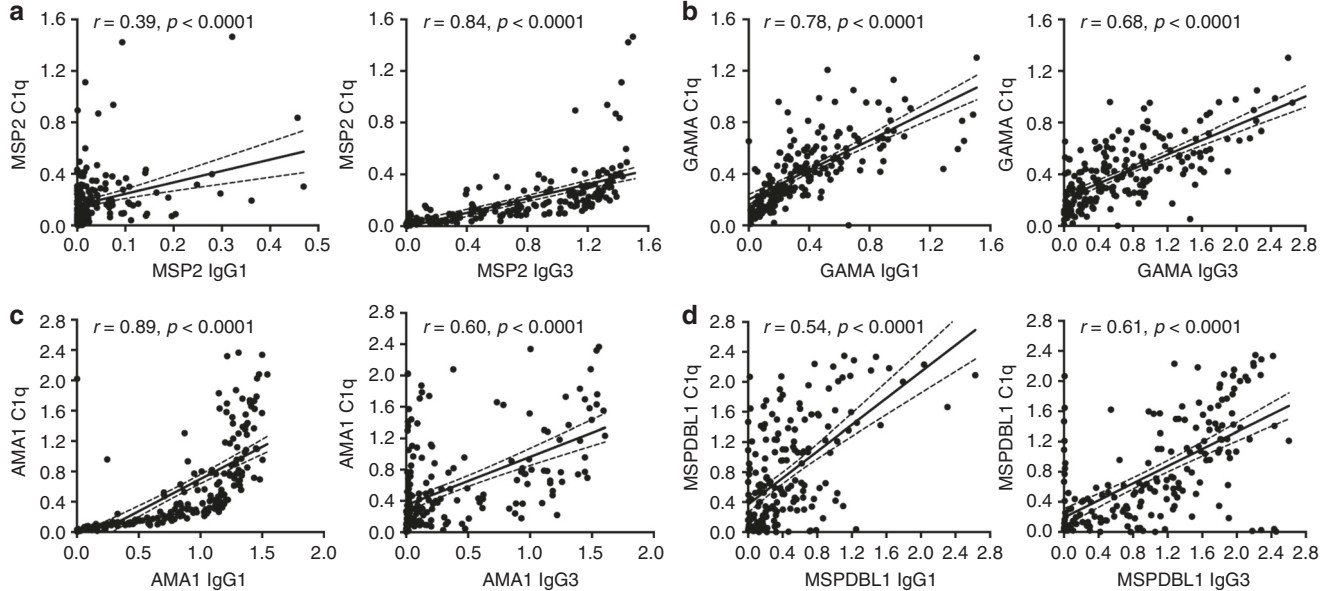

**Fig. 4** Correlations between C1q-fixation and the presence of IgG1 and IgG3 subclass antibodies against selected merozoite antigens. The correlation between the presence of antigen-specific, C1q-fixing antibodies and the presence of IgG1 or IgG3 is shown. **a** MSP2; **b** GAMA; **c** AMA1 (3D7); **d** MSPDBL1. Results are shown as OD values. Spearman's rho (r) and statistical significance (p) are indicated. Non-linear regression has been used to generate a fitted line, dotted lines are showing 95% confidence interval bands ($n = 196–201$)

complement activation pathway. We confirmed that C1q-fixation led to complement activation and down-stream formation of the MAC, and confirmed specificity by showing lack of activation when using C1q-depleted serum or a recombinant human mAb with LALA mutation in the Fc region, which ablates C1q binding. Furthermore, complement fixation on merozoites correlated with complement-dependent inhibition of invasion by antibodies. Applying these approaches in a longitudinal cohort of children acquiring immunity, we found substantial variation in the complement-fixing activity between antigens, with some antibody-antigen combinations showing limited complement-fixing ability even in the presence of high levels of total IgG. These differences may be partly explained by antibody epitope density on antigens and orientation of bound antibodies, and differences in the balance of IgG subclass responses[27,29]. Complement-fixing antibodies correlated with IgG1 and IgG3 responses, subclasses with known complement-fixation potential. These findings are consistent with the current understanding of malaria immunity where these cytophilic antibodies are associated with protective immunity[28,29]. IgM reactivity to antigens also correlated with C1q-fixation, consistent with the capacity of IgM to fix and activate complement. We found little correlation between antibody avidity (measured by SPR) and complement fixation, probably because human antibodies were already above a threshold level of antibody avidity required for complement fixation.

A striking finding was the very high protective association for complement-fixing antibodies to some antigens. There was wide variability in associations between complement-fixing antibodies and protection for different antigens tested, with some antigens having no or borderline protective associations. Interestingly, and relevant to vaccine development efforts, the strongest protective associations mostly included antigens that have not been a major focus of vaccine development to date, or are emerging vaccine candidates, including EBA140RIII-V, MSP7, RALP1, GAMA, PfRH2, PfRH5 and MSP-DBL1. While identification of potential vaccine targets has been accelerated by the increasing availability of large genomic and proteomic data sets, the lack of functional correlates of protection has been a significant hindrance for the

prioritization of vaccine candidates and the development of much-needed correlates of protection[5,50]. The plate-based complement-fixation assay optimised in this study is an important step towards addressing this gap.

Our data further highlighted specific combinations of responses that were very strongly associated with protective immunity and indicated that the breadth of antigen-specific functional antibodies is important in protective immunity. Using statistical models to estimate the PPE for all combinations of up to 6 antigens, combinations of 3 selected antigens provide near-maximal PPE, and increasing combinations to 4 or more antigens offered marginal benefit. While these findings are an encouraging step towards establishing correlates of immunity, it is important to investigate these responses in additional cohorts in future studies to understand the generalizability of our findings and to identify specific antigen combinations that could be used across different populations as a correlate of immunity. Evaluating different alleles of polymorphic antigens may also be informative. These findings also have implications for antigen inclusion in vaccine design, suggesting that specific combinations of 3–4 antigens might provide good protective immunity. Antigens that were prominent in protective combinations included RALP1, MSP7, Ripr, EBA140-RIII-V, PfRH2, MSP-DBL1 and PfRH5. These findings support further investigation of these antigens as potential vaccine candidates. Interestingly, complement-fixing antibodies to RALP1 and Ripr generally correlated only weakly or moderately with responses to other antigens suggesting that they are not strongly co-acquired with other responses. The strongest correlation with Ripr responses was with PfRH5, with which it forms a complex during invasion. PfRH5 is currently a leading vaccine candidate and our findings that complement-fixing antibodies to PfRH5 were strongly associated with protective immunity and PfRH5 was frequently present in the most protective antibody combinations (Table 2) suggest that complement-fixation should be evaluated in the ongoing clinical trials of PfRH5 vaccines, alongside other assays assessing direct antibody function to block invasion.

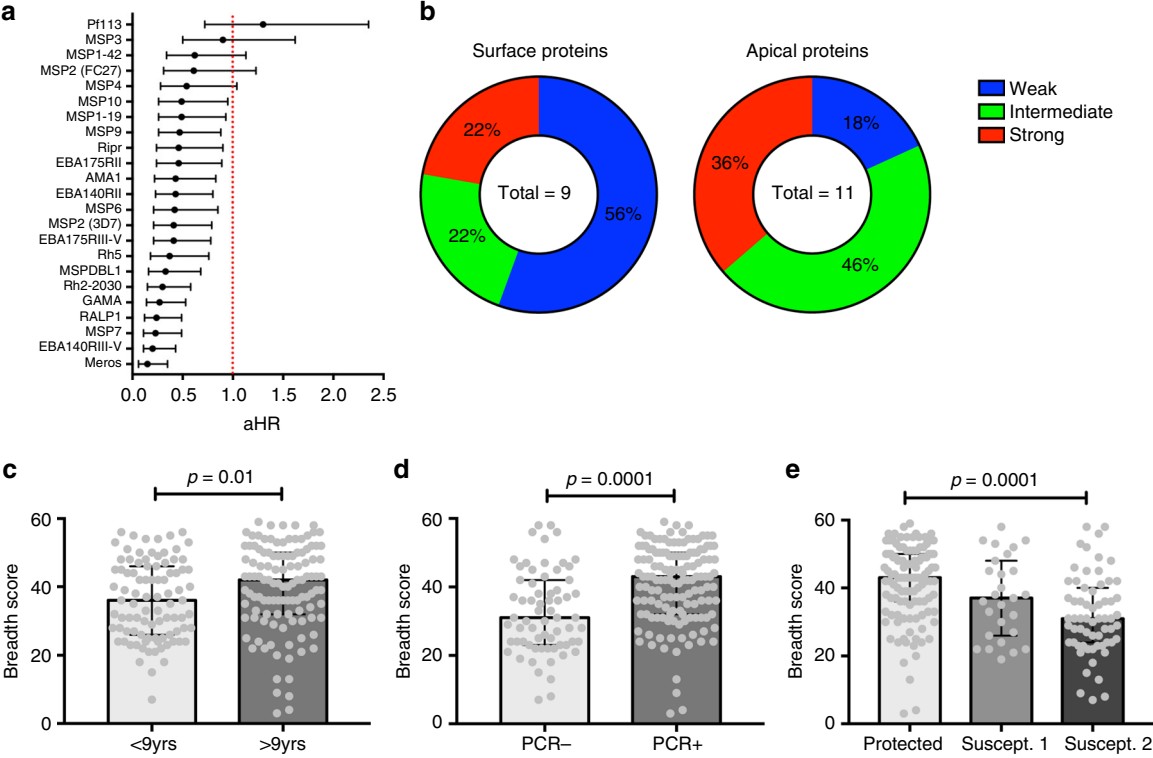

**Fig. 5** Complement-fixing antibodies against merozoite antigens are associated with protection from malaria. **a** Adjusted Hazard Ratios (aHR) for each antigen are plotted, ranked from highest to lowest based on analysis of samples and data from the Mugil cohort; the aHR for whole merozoites in also shown. aHRs were calculated using the Cox proportional hazards model, comparing high responders versus low responders, and were adjusted for confounders. Error bars indicate 95% confidence intervals. The red line indicates 0% protection (HR = 1.0). **b** Antigens were classified into tertiles according to weak ($n = 7$), intermediate ($n = 7$) and strong associations with protection ($n = 6$) from clinical malaria episodes; pie charts show the proportion of surface proteins or apical proteins associated with weak, intermediate or strong protection levels. Total $n = 20$ antigens; MSP1–42 was excluded since it is part of the same antigen as MSP1–19, and only one allele of MSP2 (3D7) was included. **c** The median breadth score of antibodies is shown stratified by age (< or >9 years of age, $n = 91$ or $n = 115$) or **d** PCR status at enrolment, PCR+ or PCR– ($n = 67$, $n = 139$). **e** shows the breadth score for all antigens tested for C1q fixation, respectively, stratified by the level of protection in the follow up period, where 'protected' is malaria-episode free during the follow up period ($n = 116$), 'susceptible 1' one malaria episode ($n = 27$) or 'susceptible 2' 2 or more malaria episodes ($n = 54$). P-values were determined by Mann-Whitney test (C + D) or Kruskal–Wallis test (E). Error bars represent the interquartile range

Prior studies have established that antibody-complement interactions can act to directly inhibit merozoite invasion and lyse merozoites through the activation of complement on the merozoite surface. Other potential mechanisms of complement may also be important for parasite clearance and immunity in vivo, such as the ability of complement to promote phagocytosis by immune cells and promote immune activation; these should be investigated in future studies. While murine models can be valuable for investigating immune mechanisms in vivo and evaluating candidate vaccines, current models have major limitations for investigating antibody-complement interactions. Murine responses to malaria infection or vaccination are predominated by IgG subclasses that do not fix complement, in contrast to the complement-fixing IgG1 and IgG3 subclasses that are predominant in human immunity[8]. Further, complement activity in laboratory mice used is ~40-times lower than humans[51]. Additionally, many of the major targets of complement-fixing antibodies that we identified are either not present in the rodent malaria species used in murine models, or orthologues differ substantially from *P. falciparum*[8]. Investment in establishing mouse or rat models of antibody-complement immunity is warranted to advance our understanding of these mechanisms in immunity and vaccine development.

In conclusion, our novel approach has addressed key gaps in knowledge of human immunity by identifying targets of complement-fixing antibodies and specific combinations of responses that are strongly linked with protective immunity to malaria. Our findings highlight several less-studied merozoite antigens as important immune targets that should be further evaluated as vaccine candidates, and supportive data for some emerging candidates. Moreover, we identified combinations of specific responses that may serve as valuable biomarkers of immunity in malaria-exposed and vaccinated individuals. The scalability of our plate-based complement-fixation assays could be applied to track differences between populations or within-populations over time, facilitating identifying populations at risk of malaria epidemics due to waning immunity[52]. Our optimised and reproducible assay provides a valuable tool with which to study the acquisition of immunity and assess vaccine efficacy. The identification of complement-fixing antibodies to specific antigens and antigen combinations provides new leads for investigation as potential vaccine targets.

## Methods

**Human subjects and samples**. Ethical approval for the use of human serum and plasma samples in these studies was obtained from the Alfred Human Research and Ethics Committee for the Burnet Institute (protocol 435.10, 327.12), the Medical Research Advisory Committee of Papua New Guinea (protocols 98.03, 05.20.01.13) and the Human Research and Ethics Committee of the Walter and Eliza Hall Institute (protocol 04.04, 06.01, 06.06). Written informed consent was obtained from all participants or, in the case of children, from their parents or guardians.

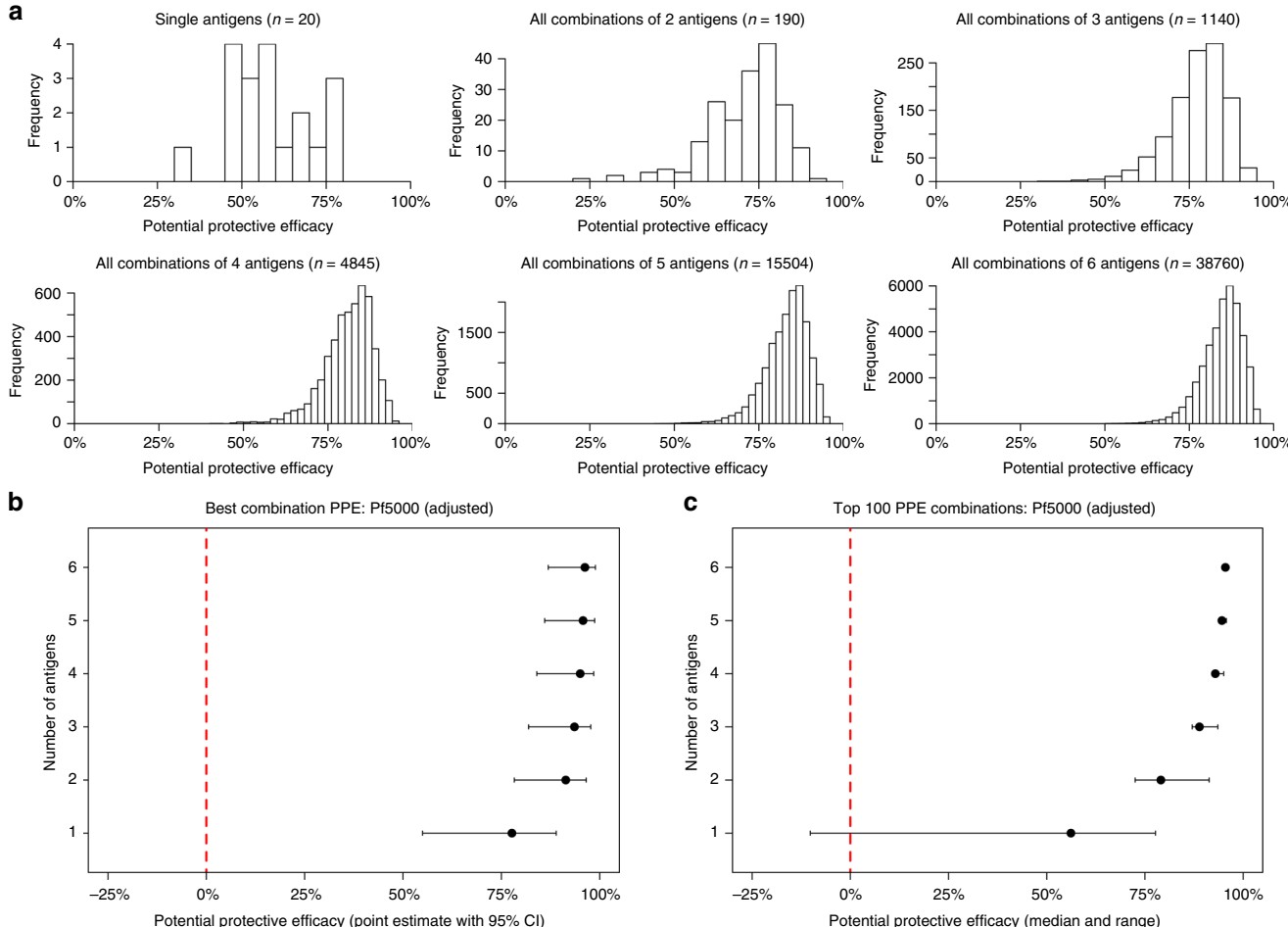

**Fig. 6** Potential Protective Efficacy (PPE) of combinations of antibody responses that fix complement. Hazard Ratios for clinical malaria episodes obtained by Cox Regression analyses were converted into Potential Protective Efficacy values (PPE = (1-aHR)*100). **a** The frequency (y-axes) of protective responses (graph 1) or combination of responses (graphs 2–6), are shown. The different levels of PPE are indicated on the x-axes. **b** shows the PPE 95% confidence interval (95% CI) of the best combination in each combination class. **c** Median PPE and range of the top 100 combinations of each combination class

**Mugil cohort**. For the longitudinal study of PNG children, plasma samples were obtained at enrolment from a prospective treatment-reinfection cohort of 206 children aged 5–14 years (median = 9.3) in Madang, PNG[42]. The prevalence of *P. falciparum* and *P. vivax*, detected by PCR, at enrolment was 67.5% and 33.9%, respectively. All children received 7 days of artesunate orally to clear any parasitaemia as described[42]. Artesunate was used because it has high efficacy and a short half-life. Children were actively reviewed every 2 weeks for symptomatic illness and parasitemia, and by passive case detection, over a period of 6 months. A clinical episode of *P. falciparum* malaria was defined as fever and *P. falciparum* parasitemia >5000 parasites/μl, and high-density parasitaemia as >5000 parasites/μl. Plasma samples used for this study were those taken at enrolment, prior to treatment with antimalarials. Sera for positive controls were obtained from adults within the Madang area. Sera for negative controls were obtained from anonymous Australian blood donors who were malaria naive.

**XMX samples**. Serum samples were collected from a cross-sectional study in Madang Province, Papua New Guinea (PNG) in 2007, and included 49 adults (median age 28 years) and 69 children (median age 6 years)[40]. These samples were included in the initial screening assays of complement fixation, and a selection of samples were used to generate data presented in Fig. 2 (selection was random, dependent on sufficient volume available).

**VT samples**. Serum samples are from a cross-sectional study and were collected from pregnant women in the second trimester, non-pregnant women, and men in Madang Province, PNG, in 2001 and 2002[53]. Samples were also collected from men accompanying children. These samples were included in the initial screening assays of complement fixation.

**Complement-fixation assays**. 96-well plates (Maxisorp, Nunc) were coated with purified merozoites at $5 \times 10^6$ merozoites per well or recombinant merozoite surface antigens at 1–2 μg/ml[24]. Briefly, plates were blocked with 1% casein, then incubated with serum samples (1/100–1/500) for 1 h, then washed with PBS-Tween 0.01%, followed by incubation for 30 min with purified C1q (10 μg/ml, Millipore) or normal serum (NS), heat-inactivated serum (HIS) from malaria naive donors, or C1q-depleted serum (Millipore) at 10% final concentration as a source of complement. After washing plates, C1q fixation was detected by incubating with rabbit anti-C1q antibodies (in-house) and then with anti-rabbit-IgG-HRP (Millipore), incubated for 1 h each. ABTS (Life Technologies) was added, incubated for 30 mins and absorbance quantified using a plate reader. All steps were conducted at room temperature (~21 °C). For studies of membrane attack complex (MAC) formation, normal serum was used as a complement source, and (rabbit-) anti-C5b/C9 antibodies (Millipore), and anti-rabbit-IgG-HRP (Millipore) were used for detection. IgG subclass ELISAs were done as previously described[29,46,54] using mouse anti-human IgG subclass monoclonal antibodies (Thermo Fischer Scientific) and anti-mouse-IgG HRP (Millipore) detection antibodies. (Detailed protocols are available from the authors on request).

**Recombinant merozoite antigens**. Proteins used in these assays were expressed recombinantly as 6-His-fusion proteins or GST-fusion proteins in *E. coli*, or in a wheat germ cell-free expression system as described previously[55,56]. Proteins MSP1-42, EBA175RII, EBA140RII and MSP4 were kindly provided by Carole Long, Annie Mo (NIH), David Narum (NIH), and Ross Coppel (Monash University), respectively. All proteins were assessed by SDS-PAGE to confirm identity and integrity, and most proteins have been validated in studies of IgG reactivity, as described[39]. Reactivity of human serum against 6-His- and GST-purification tags has previously been found to be minimal and not to impact on overall reactivity[39,46,57]. Sequences were based on the 3D7 reference sequence for all

**Table 2 Frequency of each antigen being present in the most protective combinations**

| Combination size: | Frequency of antigen in top 1% of combinations | | | |
|---|---|---|---|---|
| | 3 antigens | 4 antigens | 5 antigens | 6 antigens |
| RALP1 | 100% | 100% | 100% | 100% |
| MSP7 | 72.7% | 66.7% | 65.2% | 64.6% |
| Ripr | 36.4% | 62.5% | 85.8% | 99.2% |
| EBA140RIII-V | 18.2% | 20.8% | 16.8% | 18.1% |
| RH2-2030 | 18.2% | 25.0% | 32.9% | 39.5% |
| MSP-DBL1 | 18.2% | 18.8% | 29.7% | 37.7% |
| PfRH5 | 9.1% | 18.8% | 31.6% | 44.4% |
| GAMA | 9.1% | 18.8% | 16.8% | 18.1% |
| MSP1-19 | 9.1% | 18.8% | 19.4% | 26.4% |
| EBA175RII | 9.1% | 25.0% | 32.3% | 35.1% |
| MSP6 | 0% | 8.3% | 13.5% | 20.4% |
| EBA175RIII-V | 0% | 2.1% | 6.5% | 9.6% |
| MSP10 | 0% | 0% | 3.2% | 7.5% |
| MSP2(3D7) | 0% | 2.1% | 11.0% | 15.0% |
| MSP4 | 0% | 2.1% | 5.8% | 10.6% |
| EBA140RII | 0% | 4.2% | 12.3% | 17.8% |
| AMA1 | 0% | 2.1% | 8.4% | 14.5% |
| MSP9 | 0% | 0% | 0.6% | 1.0% |
| MSP3 | 0% | 4.2% | 8.4% | 19.9% |
| Pf113 | 0% | 0% | 0% | 0.5% |

For combinations of antigens of size $n \leq 6$, we calculated the PPE of all possible combinations. These were then ranked according to their PPE, and the top 1% of combinations with the highest PPE were analysed to identify which antigens most commonly occurred in the most protective combinations. The table shows the frequency at which each of the antigens were included in the top 1% of protective combinations

antigens; the FC27 allele of MSP2 was also used. Many antigens have minor polymorphisms that do not appear to have a major impact on antibody recognition, including MSP1-19, MSP4, MSP5, PfRH5, PfRH2, EBA140, EBA175, GAMA, PfRipr, Pf113 and RALP1. MSP2 exists as two allelic forms, and it has been previously shown that the 3D7 allele is the most prevalent in our study population[29]. We previously found that antibodies to different alleles of AMA1 are highly correlated in this population ($r = 092$, $p < 0.001$)[29]. A high prevalence of antibodies to antigens among infected children suggests that the 3D7 alleles used are representative of infections in the population[39].

**Antibody-complement invasion inhibition assays**. Invasion inhibition assays were performed with normal serum, as described previously[24]. Merozoites were purified from D10-PfPHG isolate[58] (which includes a GFP label for detection by flow cytometry), as previously described[59]. D10-PfGFP parasites were also used in previously published C1q fixation ELISA assays[24]. For complement-dependent invasion inhibition assays purified merozoite were incubated with 0.5% haematocrit RBC, 50% normal serum (from a malaria naive donor), and with 1:10 dilution of heat-inactivated sera from patient sample in duplicate as described previously[24]. Invasion was allowed to proceed for 30 min, cultures were then washed with culture medium to remove active complement, and subsequently incubated as for standard culture for 40 h. Cultures were stained with Ethidium Bromide in PBS and then analyzed via flow cytometry. Detailed methods can be found in Methods in Malaria Research, 2013[60].

**Growth inhibition assays**. GIAs were performed over two cycles of replication using validated methods[58,61]. Briefly, synchronized trophozoite stage parasites (D10-PfGFP[58]) were diluted to a final hematocrit of 1% in normal erythrocytes in culture media with a starting parasitemia of 0.1–0.3%. Parasites (25 μl) were aliquoted in 96-well U-bottom plates and sera (2.5 μl) were added at a dilution of 1/10. 5 μl of culture media was added to each well after 48 h. After 96 h, parasites were washed and stained with Ethidium Bromide in PBS and analysed by flow cytometry. All samples were tested in duplicate and expressed as an average of duplicates unless otherwise stated[60].

**Surface plasmon resonance**. Surface Plasmon Resonance (SPR) assays were performed to evaluate affinity (measured as $k_d$) as previously described[62]. Briefly, N-terminal coupling kits (Biacore) were used to bind proteins to a chip. Remaining sites were blocked by injection of 1 M ethanolamine (pH 8.5). Binding assays were performed at a flow rate of 30 μl/min at 25 °C using degassed HBS-EP running buffer. Serum samples were flowed over the bound proteins in two different dilutions, and associations were monitored for 3 min followed by a 10 min

dissociation period. Responses were monitored as a function of time and kinetic parameters ($k^a$ and $k^d$) were evaluated using the BIAevaluation 4.1 software.

**Data analysis**. Differences in complement-fixation on merozoites and specific antigens, and differences in complement fixation between groups were assessed by chi-square tests (for categorical variables) or Wilcoxon rank-sum tests (for continuous variables). Correlations between variables (continuous data) were assessed using Spearman's Rho. In graphs, fitted lines were generated using non-linear regression with dotted lines representing 95% confidence intervals.

Linear regression was performed in order to determine the association between total IgG and avidity with C1q fixation. A linear spline was applied to allow for the piecewise linear model after heteroskedasticity was detected by Breusch-Pagan/Cook-Weissberg test. A knot was determined after visual inspection of the data.

The association between antibody variables and time to first episode of clinical malaria (fever and >5000 parasites/ml) or high-density parasitemia (>5000 parasites/ml) was assessed using a Cox proportional hazards model; treatment given at enrolment to clear parasitemia facilitated a time-to-even analysis. Potential Protective Efficacy (PPE) was calculated as (1-Hazard Ratio)*100. Age and location of residence were included in Cox regression a priori as they had previously been identified as potential confounders from a range of variables in this cohort[42]. For assessment of associations between C1q fixation and protection, subjects were stratified into tertiles according to low (including those classified as 'negative'), medium, or high fixation of C1q, as determined by optical-density values for each sample. Antibody breadth scores were calculated by assigning a score of 1, 2 and 3, to low, medium and high C1q-fixing antibodies, respectively, and adding up each individual's scores for each antibody response to calculate the breadth score.

We investigated the protective associations for all possible combinations of antigen-specific complement-fixing antibodies for up to 6 antigens in a combination (from a total of $n = 20$ different antigen-specific antibody responses, $n = 190, 1140, 4845, 15504$ and $38760$ combinations, for combinations of 2, 3, 4, 5 and 6 antigens, respectively). The antibody combinations with the largest PPE were identified using Cox proportional hazards model comparing high versus low responders with first malaria episode as the outcome.

All analysis was performed with Stata/SE v13.0 Prism v7.

**Reporting summary**. Further information on experimental design is available in the Nature Research Reporting Summary linked to this article.

## Data availability

Data analysed in the preparation of this manuscript may be available from the corresponding author upon reasonable request and pending agreement from ethics committees (for clinical data). A reporting summary for this article is available as a Supplementary Information file, and the data used to generate all figures are provided as a Source Data File.

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

## Acknowledgements

RBC and human serum used for parasite culture and naive controls were provided by the Australian Red Cross Blood bank (Melbourne). We thank all participants and parents/guardians involved in the clinical studies, and the staff of the Papua New Guinea Institute for Medical Research. We thank Janine Stubbs for providing the LALA-mutant MAb. Funding was provided by the National Health and Medical Research Council of Australia (Program grant [1092789] and senior research fellowship [1077636] to J.G.B., senior research fellowship to I.M., Early Career Fellowship to M.J.B.), the International Centers for Excellence in Malaria Research (Asia-Pacific [5U19AI129392]) of the National Institutes of Health (I.M., G.B., M.T.W.), and JSPS KAKENHI(JP26253026) in Japan (T.T.). L.R., M.J.B., L.K., G.F., D.H.O., I.M. and J.G.B. are members of the NHMRC Australian Centre for Research Excellence in Malaria Elimination. The Burnet Institute acknowledges support of the NHMRC Independent Research Institutes Support Scheme and Operational Infrastructure Support of the Victorian State Government.

## Author contribution

Study design and planning: L.R., J.G.B., M.J.B. and I.M. Conducted experiments and generated reagents: L.R., M.J.B., D.W.W., G.F., R.W., H.O., K.E.M.P., T.T. and E.T. Data analysis: L.R., M.W., F.J.I.F., I.M. and J.B. Provided clinical samples and data: I.M., J.G.B., J.S.R., P.M.S. and M.J.B. Manuscript writing: L.R., J.G.B. and M.J.B. drafted the manuscript with input from all authors.

## Additional information

**Competing interests:** The authors declare no competing interests.

