## [Peer Review File · Nature Communications]

Reviewers' Comments:

Reviewer #1:

Remarks to the Author:

Review of the manuscript entitled:

“Targets of complement-fixing antibodies in protective immunity against malaria in children”

The manuscript describes the testing of a selection of recombinant malaria antigens expressed during the asexual blood stage as targets of complement fixing antibodies. Serum samples used here are derived from a cohort of children from Papua New Guinea with known malaria infection status at baseline sampling. The assay rests on the coating of antigens in ELISA plates, addition of diluted serum samples with purified complement C1q followed by detection of activated complement using anti C1q antibodies.

Major points:

First: The authors use a selection of 22 merozoite expressed antigens to assess the complement fixing activity of antigen specific antibodies of sera collected in one single cohort in Papua New Guinea. Merozoite antigens are well known for their polymorphic nature and polymorphism is one of the major immune evasion mechanisms during asexual blood stage parasitemia. The authors do not provide information on the genetic make up of the parasite populations circulating in the area from which these serum samples have been collected and how this relates to the recombinant antigen sequences tested here.

Second: The authors do not provide data on the other human IgG isotypes like IgG2 and IgG4 and seem to ignore the fact that IgM is a major isotype for complement activation. Are there data generated showing the effect of these other antibody isotypes on complement activation after binding to the recombinant antigen or purified merozoites ?

Third: The selected combination of three antigens that confer a high potential protective efficacy should be reconfirmed in a second, independent cohort. Or, the authors should clearly state in their discussion that the current findings apply to the single cohort that was studied here. Extrapolations that three antigens constitute a generalizable correlate of protection to other epidemiological settings and potentially age groups is too far fetched based on the data provided.

Forth: The materials and methods section does not allow for the repetition of the presented experiments. Most importantly, the used sources of purified complement and complement detecting antibodies used in the ELISA are not provided. Also, the strain from which the merozoites are purified and incorporated into the assays remain unclear to the reader.

Other points:

In general, the manuscript does not follow the format requested by Nature Communications and the literature cited seems to be sometimes redundant.

Abstract: word count should be 150 words.

The authors claim that this assay presented in the study (line 58) is high through put for assessing the activity of anti-merozoite antibodies fixing complement might be misleading. This assay still relays on the ELISA format and the availability of recombinant expressed antigens.

The sentence in the abstract, line 70, that a side by side comparison of the growth inhibition assay (GIA) with the complement fixing antibody assay targeting either the whole merozoite or distinct

recombinant merozoite expressed antigens has not been demonstrated to the understanding of this reviewer.

Ethical approval:

are there project and approval numbers by the different ethical committees that could be presented ?

Cohorts:

Why were the members of the Mugil cohort treated with 7 days of artesunate monotherapy ?

The samples of the XMX and VT cohorts: in which assays have these samples been used ?

In general, it would be helpful to have the number of samples analysed (n) clearly provided in the Figure legends. This applies also to the statistical analyses conducted throughout the paper.

In Figure 1, the x-axis scale and meaning remains unclear to the reader. The selection of serum samples assessed in the assays used in Figure 1B, C and D remain unexplained.

In Figure 1A, is this a side by side comparison of the two different growth inhibition assays at the same day and using similar reagents or is this a comparison between historical data and novel data ?

Line 96: what kind of immunity is meant in this sentence ?

Line 119, following citation is missing: Zenklusen et al., JID, 2018

Lines 262 – 266: repetition of introduction, could be removed

Line 341: on what basis were these serum samples selected ? On availability, on previous results or a scientific rationale ?

Figure 2 A, B and C: what are the x and y axis showing ? OD values ?

Line 381 - 382: This sentence seems to overrate the data presented before in the manuscript

Figure legend to Table 1 is incomplete, only 18 antigens of the 23 antigens mentioned in the table are explained in terms of volunteers analysed, please check.

Figure 4A: why are the whole merozoites as target of the antibodies have a negative aHR ?

Line 541: sentence is unclear

Line 580: please provide formula as to how the possible combinations have been calculated

Table 3: what are these percentage values of the individual antigens shown ? Prevalence in the cohort tested ?

Reviewer #2:

Remarks to the Author:

This study demonstrates the importance of complement-fixing antibodies in protective immunity to malaria by preventing merozoite invasion into red blood cells. This study showed side-by-side comparisons between widely used in vitro assays that heat-inactivate sera with assays that retain complement fixation functionality. In addition, this study evaluates the Potential Protective Efficacy of a panel of merozoite surface and apical proteins showing a ranked combination of responses to 3-5 antigens as sufficient for protection, which is reassuring for malaria vaccine development strategies. This study suggests MSP7, RALP1 and Ripr may be important immune targets conferring antibody mediated protection against malaria and could be investigated as new malaria vaccine candidates. However, the authors did not fully discuss the implications for including complement dependent screening methods for malaria vaccine candidate selection along with direct methods that may find other antigens that confer protection to be included in a combination vaccine design. The manuscript could be further improved by clarifying experimental details used and comparisons of their findings in relation to other published studies. The analysis was appropriate, and methods appropriately validated. In summary, this study is novel and highly relevant to the scientific community.

Major comments:

- 1) The Abstract is poorly written, redundant and vague. It would benefit from a severe editing and including more conclusive statements that highlight the specific results from this study.
- 2) The Results were concise and by and large well presented. However, it would be helpful if you were consistent with your nomenclature when referring to the various methods. Your previous study referred to "Ab-C' IIA" and 'Ab-C' GIA' compared to "direct" IIA and GIA. But this manuscript is not even internally consistent in labeling these assays from the text to the figures making it cumbersome to follow each comparison. Referring to them "complement dependent" might be preferred by your group going forward but then it would be helpful if the figures in this manuscript were labeled throughout with mechanistic modifiers, as such "CD-IIA, CD-GIA, Direct-IIA, Direct-GIA", instead of 'standard' or 'traditional'. The most egregious example is Figure 1.
- 3) Figure 2 could be better labeled indicating on the Figure itself that graph 2B are results for MSP2. As for Figures 2C and 2D, the experimental comparisons could be more clearly labeled as ELISA results measuring C1q fixation and MAC formation, respectively (with OD value as the unit of measure for each of the different ELISA assays).
- 4) Table 1 – is there a more concise way to report the varying sample sizes for each antigen, or to only include samples with complete results for all antigens (n=87, 107, 64 and 130)? It would seem based on your p-values that omitting those few samples would not change the significance of your findings.
- 5) Table 2 – a heatmap legend is missing that shows colors as levels otherwise this is an informative way to show this type of data. If the p-value is not significant, then the box could be clear with no color.
- 6) Ripr and RALP1 were ranked highly for the combination of antibodies that predict protection (Table 3) yet they had a low prevalence of C1q fixing antibodies and did not correlate with other complement fixing antibodies (Table 1 and Table 2, respectively). This was not reconciled in your concluding statements in the Discussion.
- 7) The Discussion section of the manuscript could be edited to reduce redundancies with methods explained in detail above. The second and third paragraphs could be omitted unless they include discussions of your results in more detail within the context of other published studies, which also found IgG1 and IgG3 to be important for protection against malaria. How does your CD assay results confirm or refute our understanding of humoral immunity to different malaria antigens?

Response to Reviewers

General comments:

We thank the reviewers for their helpful suggestions and we have addressed all the points raised in our revised manuscript, and the revisions are outlined below. In addition to addressing these comments, we have also carefully reviewed the manuscript throughout to ensure clarity and reduce redundancy and repetition, and comply with Nature Communications style and formatting.

We have uploaded a 'Marked-up' copy of our manuscript to facilitate the review of changes made.

We have also included a data file with our submission, a new requirement of the journal, which includes the source data used to plot all our figures. A large amount of additional source data, summary variables, and descriptive data are provided in the Supplementary Material.

Reviewer #1:

The manuscript describes the testing of a selection of recombinant malaria antigens expressed during the asexual blood stage as targets of complement fixing antibodies. Serum samples used here are derived from a cohort of children from Papua New Guinea with known malaria infection status at baseline sampling. The assay rests on the coating of antigens in ELISA plates, addition of diluted serum samples with purified complement C1q followed by detection of activated complement using anti C1q antibodies.

Response: We thank the reviewer for the helpful feedback on our manuscript. We have addressed all points raised as outlined below.

Major points:

First: The authors use a selection of 22 merozoite expressed antigens to assess the complement fixing activity of antigen specific antibodies of sera collected in one single cohort in Papua New Guinea. Merozoite antigens are well known for their polymorphic nature and polymorphism is one of the major immune evasion mechanisms during asexual blood stage parasitemia. The authors do not provide information on the genetic make-up of the parasite populations circulating in the area from which these serum samples have been collected and how this relates to the recombinant antigen sequences tested here.

Response: The issue of polymorphisms in some merozoite antigens is an important point to clarify in the revised manuscript. We have now added some explanation and clarification of this in the methods section (line 555) and we have added some additional analysis in the results section (line 302). There are several points to consider. Detailed data on the circulating strains for all merozoite antigens is not available, and there is a limited understanding of how polymorphisms impact on antibody binding. Therefore it is difficult to extrapolate parasite genetic data to immunologic reactivity and specificity. All the antigens we have used are derived

from the 3D7 reference genome which has been used in all published studies that evaluate antibodies to multiple antigens, so this is a widely accepted approach. We show in our paper (and report previously), that antibodies to merozoite antigens of the 3D7 reference genome are commonly recognised by malaria exposed individuals in our study population such that there is a high prevalence of antibodies among malaria-infected children in our study population (see Table S1, and Richards JS et al, 2013). A further consideration is that some antigens are not polymorphic and others have only minor polymorphisms that do not appear to have a major impact on antibody recognition. This includes the following antigens: MSP1-19, MSP4, MSP5, PfRH5, PfRH2, EBA140, EBA175, GAMA, PfRipr, P113, and RALP1. Polymorphisms in AMA1 can impact on antibody binding to some extent, and we have previously established that infections with 3D7-like alleles are common in our study population (Terheggen U et al, 2014), and antibodies to different AMA1 alleles were highly correlated ($r=0.92$; Stanisic et al, 2009). MSP2 exists as two allelic forms, and it has been previously shown that the 3D7 allele is the most prevalent in our study population (Stanisic DI et al, 2009). Naturally-acquired antibodies to MSP2 are largely allele-specific (Feng G, et al JID 2018). Among the antigens we evaluated, polymorphism and allele-specificity of antibodies is most significant for MSP2. Therefore, the revised manuscript we have added some text to the results section noting this issue and we added some additional analysis of protective associations for combined responses to the two MSP2 alleles (from line 302). These analyses showed that high levels of complement-fixing antibodies to both alleles had a strong protective association. We have also added a note in the discussion about future studies evaluating polymorphisms.

Second: The authors do not provide data on the other human IgG isotypes like IgG2 and IgG4 and seem to ignore the fact that IgM is a major isotype for complement activation. Are there data generated showing the effect of these other antibody isotypes on complement activation after binding to the recombinant antigen or purified merozoites?

Response: Of the IgG subclasses, only IgG1 and IgG3 can effectively fix and activate complement via the classical pathway. IgG2 and IgG4 have little or no complement fixing activity (E.g. see review by Irani V et al, Molec Immunol 2015). This is why our correlation analysis is focused on IgG1 and IgG3 only. Furthermore, it has been previously demonstrated that IgG1 and IgG3 are the dominant responses to merozoite antigens in this population, with very little IgG2 and IgG4 (as has been reported from many malaria-endemic populations. In the revised manuscript we have added some text to the results section to explain these two points (with relevant references)(from line 253).

Regarding IgM, we agree with the reviewer that we should include some consideration of IgM given its potential to fix and activate complement. We have now included some text in the results indicating correlations between IgM responses and complement fixation for selected antigens as examples (from line 264).

Third: The selected combination of three antigens that confer a high potential protective efficacy should be reconfirmed in a second, independent cohort. Or, the authors should clearly state in their discussion that the current findings apply to the single cohort that was studied here. Extrapolations that three antigens constitute a generalizable correlate of protection to other epidemiological settings and potentially age groups is too far fetched based on the data provided.

Response: We agree with reviewer that it is important to investigate responses in additional cohorts in future studies to understand the generalizability of our findings

and to identify specific antigen combinations that could be used across different populations as a correlate of immunity. Equally, we think our manuscript reports exciting new findings on functional correlates of protection that are valuable in understanding immunity to inform vaccine and biomarker development. In our original manuscript we had tried to be careful not to overstate our findings on this point. In the revised manuscript we have reviewed our wording and made edits where necessary. In the discussion, we have now specifically indicated the need for future studies in different populations to evaluate complement fixing antibodies to specific antigens and combinations as correlates of protection (from line 449). We hope that the reviewer and editors appreciate that conducting studies of this size in additional cohorts would be a very major undertaking and is beyond the scope of this manuscript. It would require the expression and validation of all merozoite antigens to be used, all the optimisation work needed for the assays in the study cohort, conducting all assays with multiple merozoite antigens, and the extensive analysis involving multiple different responses. We are in discussions with other malaria researchers who may be interested in applying these assays, and we have offered to provide reference reagents and detailed protocols for such studies. We believe the findings in our present manuscript do provide some exciting new insights and approaches than can be applied in future studies to better understand human immunity and work towards the establishment of much-needed correlates of protection.

Fourth: The materials and methods section does not allow for the repetition of the presented experiments. Most importantly, the used sources of purified complement and complement detecting antibodies used in the ELISA are not provided. Also, the strain from which the merozoites are purified and incorporated into the assays remain unclear to the reader.

Response: Clarifications have now been added to the materials and methods section, including all reagent sources and strains of parasite antigens and cultures. We have noted in the text that a detailed protocol is available from the authors on request. Furthermore, we have offered to provide reference reagents (positive and negative controls, and validated detection reagents) to other researchers to help them establish these assays.

Other points:

In general, the manuscript does not follow the format requested by Nature Communications and the literature cited seems to be sometimes redundant.

Response: Nature Communications does not require strict adherence to their published format for initial submissions. In our revised manuscript we have now taken care to adhere to all publication guidelines, after the editor provided us with a checklist.

Abstract: word count should be 150 words.

Response: Abstract has been edited to conform with Nature Communications guidelines.

The authors claim that this assay presented in the study (line 58) is high throughput for assessing the activity of anti-merozoite antibodies fixing complement might be misleading. This assay still relies on the ELISA format and the availability of recombinant expressed antigens.

Response: References to high throughput nature of developed assay have been removed from the manuscript. Instead we have noted that the assay is an efficient approach. We believe the methods can be adapted to high throughput platforms, but this is yet to be formally validated. We are in discussions about these approaches with collaborators.

The sentence in the abstract, line 70, that a side by side comparison of the growth inhibition assay (GIA) with the complement fixing antibody assay targeting either the whole merozoite or distinct recombinant merozoite expressed antigens has not been demonstrated to the understanding of this reviewer.

Response: The abstract has been re-written in the revised manuscript following feedback from the reviewers and to comply with the journal's format requirement of 150 words. Edits have been made to the results, figure legends and discussion sections to ensure that the type of analyses and the samples and data included in analyses are clear. In Fig 1A we have compared the protective associations (expressed as Hazard Ratios generated using Cox Proportional Hazards models) for complement-fixing antibodies and growth-inhibitory antibodies (the same parasite isolate and was used in both assays, and the analyses include the same cohort subjects). In Fig 1B, C, and D, we provide further comparisons between different antibody activities from further testing of a subset of samples. In Fig 4 (and supplementary Table), we show the protective associations for antigen-specific complement-fixing antibodies and whole merozoites in the same cohort used in Figure 1. We hope that this is now clear and the reworded abstract addresses this point.

Ethical approval: are there project and approval numbers by the different ethical committees that could be presented?

Response: These have now been added.

Cohorts: Why were the members of the Mugil cohort treated with 7 days of artesunate monotherapy?

Response: It was considered appropriate/ideal (for ethical and health reasons) to treat children for parasitemia when they were tested at enrolment. Treatment to clear any blood-stage parasitemia prior to follow-up also enabled the analysis to be based on a time-to-event approach. Artesunate was used because it has high efficacy and a short half-life and is included in the national guidelines as an accepted treatment. Some details have now been added to explain this in the Methods section (lines 509, 610).

The samples of the XMX and VT cohorts: in which assays have these samples been used?

Response: XMX cohorts were used in data presented in Figure 2. This is now indicated in the Figure legend. Further, these samples and VT samples were used in preliminary screening of antigens prior to the extensive testing in the longitudinal children cohort. This is now explained in the methods.

In general, it would be helpful to have the number of samples analysed (n) clearly provided in the Figure legends. This applies also to the statistical analyses conducted throughout the paper.

Response: Sample numbers have been added to figure legends and table footnotes.

In Figure 1, the x-axis scale and meaning remains unclear to the reader. The selection of serum samples assessed in the assays used in Figure 1B, C and D remain unexplained.

Response: The Figure legend has been modified for clarity to provide details of Figure 1A analysis, and sera samples used in subsequent figures.

In Figure 1A, is this a side by side comparison of the two different growth inhibition assays at the same day and using similar reagents or is this a comparison between historical data and novel data ?

Response: Figure 1A is a comparison of the protective associations of functional antibodies measured in growth inhibition assays that are performed in complement free conditions (new data included in this paper), and functional antibodies that fix C1q to the merozoite surface (using previously published data from Boyle et al, 2015, Immunity – this was noted in the manuscript). Assays were performed in the same longitudinal cohort of children and using the same parasite isolate. Edits have been made to the results text for clarity.

Line 96: what kind of immunity is meant in this sentence?

Response: This sentence has been clarified to indicate protective immunity from malaria disease.

Line 119, following citation is missing: Zenklusen et al., JID, 2018

Response: This reference is now included.

Lines 262 – 266: repetition of introduction, could be removed

Response: This section has been modified to address this point.

Line 341: on what basis were these serum samples selected ? On availability, on previous results or a scientific rationale ?

Response: Samples were selected randomly. This clarification is now included in the Methods text (line 522).

Figure 2 A, B and C: what are the x and y axis showing ? OD values ?

Response: This has now been clarified in the figure legend. The data are presented as OD values.

Line 381 - 382: This sentence seems to overrate the data presented before in the manuscript

Response: The wording of this section has now been revised.

Figure legend to Table 1 is incomplete, only 18 antigens of the 23 antigens mentioned in the table are explained in terms of volunteers analysed, please check.

Response: The figure legend is now complete. As highlighted by the reviewer, numbers for antigens Pf113, Rh2-2030, MSP9 and MSP10 were missing in the original submission and have now been added in the revised manuscript.

Figure 4A: why are the whole merozoites as target of the antibodies have a negative aHR ?

Response: We are not clear regarding the meaning of this question. In Fig 4A, the aHR for whole merozoites is not negative. The aHR for C1q fixing antibodies to whole merozoites is 0.15 (95%CI 0.06-0.35). All the aHR values with 95% CI limits are reported in the Supplementary file, Table S3

Line 541: sentence is unclear

Response: This sentence has been modified for clarity and now reads: Total n=20; MSP1-42 was excluded since it is part of the same antigen as MSP1-19, and only on allele of MSP2 (3D7) was included. (from line 345)

Line 580: please provide formula as to how the possible combinations have been calculated

Response: The numbers of combinations arising from having combinations of 2,3,4,5, or 6 antigens (n=20 antigens in total) was determined using standard mathematical calculations for combinations. We have addressed the wording to make this clearer.

Table 3: what are these percentage values of the individual antigens shown? Prevalence in the cohort tested?

Response: Table 3 values are the frequency at which each antigen occurs in the top 1% of antigen combinations ranked by protective efficacy. For example, of the total possible 3 antigen combinations (1140), the most protective 11 combinations all had RALP1 (100% in the table). This was explained in the footnote to the table. We have now made some revisions to make this clearer. This section of the results has also been modified to improve the clarity (Paragraph starting at line 365).

Reviewer #2:

This study demonstrates the importance of complement-fixing antibodies in protective immunity to malaria by preventing merozoite invasion into red blood cells. This study showed side-by-side comparisons between widely used in vitro assays that heat-inactivate sera with assays that retain complement fixation functionality. In addition, this study evaluates the Potential Protective Efficacy of a panel of merozoite surface and apical proteins showing a ranked combination of responses to 3-5 antigens as sufficient for protection, which is reassuring for malaria vaccine development strategies. This study suggests MSP7, RALP1 and Ripr may be important immune targets conferring antibody mediated protection against malaria and could be investigated as new malaria vaccine candidates.

However, the authors did not fully discuss the implications for including complement dependent screening methods for malaria vaccine candidate selection along with direct methods that may find other antigens that confer protection to be included in a combination vaccine design.

The manuscript could be further improved by clarifying experimental details used and comparisons of their findings in relation to other published studies.

The analysis was appropriate, and methods appropriately validated. In summary, this study is novel and highly relevant to the scientific community.

Response: We thank the reviewer for constructive feedback on our manuscript. We have addressed all concerns as outlined below.

Further information has been added to the methods section to clarify experimental details. The Discussion has been edited to contextualise the implications of our findings in vaccine development.

Major comments:

1) The Abstract is poorly written, redundant and vague. It would benefit from a severe editing and including more conclusive statements that highlight the specific results from this study.

Response: The abstract has been significantly edited to improve clarity and comply with the journal's limit of 150 words.

2) The Results were concise and by and large well presented. However, it would be helpful if you were consistent with your nomenclature when referring to the various methods. Your previous study referred to "Ab-C' IIA" and 'Ab-C' GIA' compared to "direct" IIA and GIA. But this manuscript is not even internally consistent in labeling these assays from the text to the figures making it cumbersome to follow each comparison. Referring to them "complement dependent" might be preferred by your group going forward but then it would be helpful if the figures in this manuscript were labeled throughout with mechanistic modifiers, as such "CD-IIA, CD-GIA, Direct-IIA, Direct-GIA", instead of 'standard' or 'traditional'. The most egregious example is Figure 1.

Response: Clarifications have been made throughout the manuscript in order to increase clarity. All GIA are performed in complement free conditions (that is measuring direct activity of antibodies). All Invasion assays used in this manuscript included complement and are now indicated as antibody-complement invasion inhibition assays (AbC-IIA), which is more consistent with our previous publication.

We agree that the legend to Figure 1 was not sufficiently clear and we have now edited that to improve the clarity.

3) *Figure 2 could be better labeled indicating on the Figure itself that graph 2B are results for MSP2. As for Figures 2C and 2D, the experimental comparisons could be more clearly labeled as ELISA results measuring C1q fixation and MAC formation, respectively (with OD value as the unit of measure for each of the different ELISA assays)*

Response: We agree that the original figure legend was not very clear. This has now substantially re-written to make it clearer.

4) *Table 1 – is there a more concise way to report the varying sample sizes for each antigen, or to only include samples with complete results for all antigens (n=87, 107, 64 and 130)? It would seem based on your p-values that omitting those few samples would not change the significance of your findings.*

Response: We believe the most transparent presentation of the data is to list all the sample sizes in the footnote. In the published format the footnote will appear more concisely

5) *Table 2 – a heatmap legend is missing that shows colors as levels otherwise this is an informative way to show this type of data. If the p-value is not significant, then the box could be clear with no color.*

Response: Table 2 has been modified to provide a colour guide for the strength of correlations which makes it easier to interpret. We have indicated levels of statistical significance in the Table footnote

6) *Ripr and RALP1 were ranked highly for the combination of antibodies that predict protection (Table 3) yet they had a low prevalence of C1q fixing antibodies and did not correlate with other complement fixing antibodies (Table 1 and Table 2, respectively). This was not reconciled in your concluding statements in the Discussion.*

Response: The prevalence of C1q fixing antibodies to Ripr and RALP1 are relatively high at 66.2% and 84.7% respectively; furthermore, among kids with active infection (PCR+), the prevalences were higher at 88.6%. These data are presented in Supplementary Table S1. Table 1 presents the magnitude (or levels) of the antibody response. In the first submission the labelling of Table 1 was not as clear as it could have been and it has now been modified to ensure clarity. We agree that it is interesting that the correlations between RALP1 and Ripr with other antigens was relatively low for several comparisons. The correlations between C1q fixing antibodies to Ripr and other antigens were generally either low or non-significant. Of interest, the strongest correlation for Ripr was with PfRH5, with which it forms a complex during invasion. The correlations between RALP1 and other antigens were more evident, and four antigen correlations had $\rho > 0.5$ (MSP9, EBA140-RII-V, EBA175 RII-V, and GAMA). We have now briefly commented on this in the discussion (from line 458). On the other hand, both RALP1 and Ripr did show protective associations and both antigens featured in the most protective antigen combinations, especially RALP1 (Table 3). As suggested, we have now briefly commented on these points in the discussion.

7) *The Discussion section of the manuscript could be edited to reduce redundancies with methods explained in detail above. The second and third paragraphs could be omitted unless they include discussions of your results in more detailed within the*

context of other published studies, which also found IgG1 and IgG3 to be important for protection against malaria. How does your CD assay results confirm or refute our understanding of humoral immunity to different malaria antigens?

Response: The Discussion has been edited to reduce redundancies and make it more concise. Our findings advance our understanding of immunity to malaria by providing the first data on functional antibodies to multiple antigens and their association with protection. We have considered a number of points in the Discussion section on the implications of these findings for the current understanding of immunity, the implications for vaccine development, and evaluation of vaccine candidates in pre-clinical studies and clinical trials. We have made revisions to the Discussion section to bring out these points more clearly.

Reviewers' Comments:

Reviewer #1:

Remarks to the Author:

the authors have addressed the questions and issues raised by this reviewer to its satisfaction.